# A modular biomimetic strategy for the synthesis of macrolide P-glycoprotein inhibitors via Rh-catalyzed C-H activation

Lu Chen[1,2,5], Haitian Quan[1,2,5], Zhongliang Xu[1,2], Hao Wang[1,2], Yuanzhi Xia[3], Liguang Lou[1,2✉] & Weibo Yang[1,2,4✉]

One of the key challenges to overcome multidrug resistance (MDR) in cancer is the development of more effective and general strategies to discover bioactive scaffolds. Inspired by natural products, we describe a strategy to achieve this goal by modular biomimetic synthesis of scaffolds of (Z)-allylic-supported macrolides. Herein, an Rh(III)-catalyzed native carboxylic acid-directed and solvent-free C−H activation allylation with high stereoselectivity and chemoselectivity is achieved. The generated poly-substituted allylic alcohol as a multifunctional and biomimetic building block is crucial for the synthesis of (Z)-allylic-supported macrolides. Moreover, the unique allylic-supported macrolides significantly potentiate the sensitivity of tumor cells to cytotoxic agents such as vinorelbine and doxetaxel by reversing p170-glycoprotein-mediated MDR. Our findings will inspire the evolution of synthetic chemistry and open avenues for expedient and diversified synthesis of bioactive macrocyclic molecules.

---

[1] Chinese Academy of Sciences Key Laboratory of Receptor Research, Shanghai Institute of Materia Medica (SIMM), Chinese Academy of Sciences, Shanghai, China. [2] University of Chinese Academy of Sciences, Beijing 100049, China. [3] College of Chemistry and Materials Engineering, Wenzhou University, Wenzhou 325035, China. [4] Key Laboratory for Functional Material, Educational Department of Liaoning Province, University of Science and Technology Liaoning, Anshan 114051, China. [5] These authors contributed equally: Lu Chen, Haitian Quan. ✉email: lglou@simm.ac.cn; yweibo@simm.ac.cn

Many natural macrocyclic small molecules based on unique scaffolds have served as an inspiration and resource in drug discovery, or been harnessed as probes for targets validation in chemical biology[1]. For instance, the drug for Cushing's disease Pasireotide[2], Histone Deacetylase (HDAC) inhibitor Romidepsin[3] and cGAS-STING inhibitor Astin C[4] (Fig. 1a). It is notable that most of the natural macrocyclic small molecules are encoded with fundamental building blocks, such as natural amino acids, at a proper position. Since these building blocks inherently possess structural biology information, the natural macrocyclic molecules derived from them could exert multifunctional biological activities when they interact with different targets[5–9]. Although they possess valuable characteristics, gene expression limitation of microbes or plants and difficulty of resupply could limit both the number and types of macrocyclic compounds accessible.

With these issues in mind, we want to develop a short and modular biomimetic strategy, which simply utilizes the fundamental building blocks from living organism's endogenous ligand, such as amino acids, to expeditiously access and enrich diverse natural macrocycle-like chemical space. Macrolides could provide an excellent opportunity for validating this strategy, as they are prevalent across all major natural molecules[10]. Inspired by NPs and our interest in construction of scaffold-diverse libraries[11–14], we set out to create a series of allylic-supported macrolides via an orchestrated assembly of readily available carboxylic acids, natural amino acids and vinylethylene carbonates[15] using biomimetic modularization strategy (Fig. 1b). A retrosynthetic analysis indicated that a successive and concise C–H allylation, amidation, and esterification could translate these building blocks faithfully into the target molecules. However, several challenging issues on native carboxylic acid-directed C–H activation allylation[16,17] still need to be addressed: First, chemoselectivity between C–C bond and C–O bond formation should be a dilemma. While an amidation of carboxylic acid could avoid the C–O allylation product and direct the C–H bond activation C–C allylations[18], an inevitable drawback is the need for additional synthetic steps for installation and removal of the directing group. Second,

controlling stereoselectivity of poly-substituted allylic alcohols remained a formidable challenge. Third, the branch/linear regioselectivity represented a common issue (Fig. 1c).

Herein, we realize these ideas by Rh (III)-catalyzed C–H allylation of carboxylic acids and report a biomimetic modularization strategy, which utilizes the readily available building blocks to assemble allylic-supported macrolides that would be difficult or impossible to obtain by other methods. Importantly, these allylic-supported macrolides successfully transfer pharmacologically relevant features and efficiently surmount P-glycoprotein(P-gp)-mediated multidrug resistance (MDR) in cancer chemotherapy.

## Results

**Reaction optimization.** Stimulated by the aforementioned challenges, we commenced our building blocks assembly by optimizing C–C bond formation (Table 1). The readily available 2-methylbenzoic acid **1a** and 4-phenyl-4-vinyl-1,3-dioxolan-2-one **2a** were chosen as the model substrates. When **1a** and **2a** were treated with a low valent $Pd_2(dba)_3 \cdot CHCl_3$ catalyst in dimethylformamide (DMF) at 60 °C, an exclusive C–O bond instead of C–C bond formation product **4aa** was obtained in 45% NMR yield. Interestingly, replacement of low valent $Pd_2(dba)_3 \cdot CHCl_3$ catalyst to high valent Rh (III) catalyst dramatically changed the outcome of chemoselectivity, solely affording C–C bond formation product **3aa** in 36% NMR yield albeit with low stereoselectivity. The Z-geometry configuration of **3aa** was confirmed by NOE analysis. The switchable chemoselectivity in this reaction could be attributed to the nature of controlled reaction intermediates, namely rhodacycle and π-allyl Pd intermediates[19]. Encouraged by these results, we next investigated different solvents and bases. It was found that the reaction was not compatible with 1,2-dichloroethane (DCE) or toluene and no desired product **3aa** could be afforded. Further screening of various bases revealed that $K_2CO_3$ was the optimal base, whereas other bases gave inferior results. Meanwhile, the amounts of the $K_2CO_3$ was also optimized and the excess $K_2CO_3$ is not likely beneficial for **3aa** formation. In contrast, the use of 0.5 equiv. $K_2CO_3$ led to a

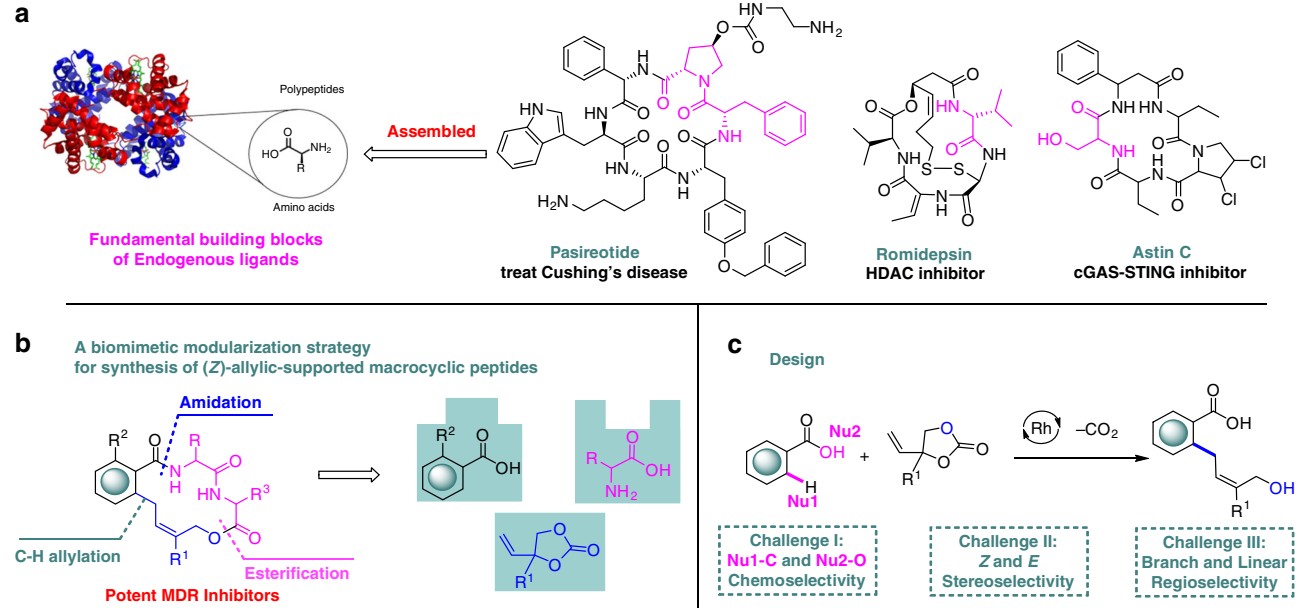

**Fig. 1 Our design to create functional macrolides by a biomimetic modularization strategy. a** Natural macrolides and macrocyclic peptides assembled by fundamental building blocks. **b** A biomimetic modularization strategy for synthesis of (Z)-allylic-supported macrocyclic peptides. **c** The design of Rh (III)-catalyzed C–H allylation of carboxylic acids. MDR: multidrug resistance, HDAC: histone deacetylase, cGAS: cyclic GMP-AMP synthase, STING: stimulator of interferon genes, Nu: nucleophile.

**Table 1 Optimization of building blocks assembly by chemoselective formation of C–C bond.**

| Entry[a] | Catalyst | Ligand | Base | Solvent | Yield(%)[b] | |
|---|---|---|---|---|---|---|
| | | | (equiv.) | | 3aa | 4aa |
| 1 | Pd$_2$(dba)$_3$•CHCl$_3$ | PPh$_3$ | – | CHCl$_3$ | N.D. | 45 |
| 2 | [Cp*RhCl$_2$]$_2$/AgSbF$_6$ | – | Cs$_2$CO$_3$ (2.0) | CHCl$_3$ | 36 (1:1) | N.D. |
| 3 | [Cp*RhCl$_2$]$_2$/AgSbF$_6$ | – | Cs$_2$CO$_3$ (2.0) | DCE | N.D. | N.D. |
| 4 | [Cp*RhCl$_2$]$_2$/AgSbF$_6$ | – | Cs$_2$CO$_3$ (2.0) | toluene | N.D. | N.D. |
| 5 | [Cp*RhCl$_2$]$_2$/AgSbF$_6$ | – | CsOAc (2.0) | CHCl$_3$ | 21 (2:1) | N.D. |
| 6 | [Cp*RhCl$_2$]$_2$/AgSbF$_6$ | – | K$_2$CO$_3$ (2.0) | CHCl$_3$ | 40 (1:1) | N.D. |
| 7[c] | [Cp*RhCl$_2$]$_2$/AgSbF$_6$ | – | K$_2$CO$_3$ (2.0) | CHCl$_3$ | 34 (2:1) | N.D. |
| 8[d] | [Cp*RhCl$_2$]$_2$/AgSbF$_6$ | – | K$_2$CO$_3$ (2.0) | – | 46 (>20:1) | N.D. |
| 9[d] | [Cp*RhCl$_2$]$_2$/AgSbF$_6$ | – | K$_2$CO$_3$ (1.0) | – | 56 (>20:1) | N.D. |
| 10[d] | [Cp*RhCl$_2$]$_2$/AgSbF$_6$ | – | K$_2$CO$_3$ (0.5) | – | 79 (>20:1) | N.D. |
| 11 | [Cp*RhCl$_2$]$_2$/AgSbF$_6$ | – | — | – | N.D | N.D |

[a] Reaction conditions: **1a** (0.05 mmol), **2a** (0.075 mmol), catalyst (5 mol %), ligand (20 mol %), AgSbF$_6$ (20 mol%), solvent (0.5 mL), 60 °C, 12 h. After the reaction was complete,1 mL DMF, 0.15 mmol K$_2$CO$_3$ and 0.3 mmol MeI were added and the mixture was stirred at rt for 3 h.
[b] Yields were determined by the $^1$H NMR using CH$_2$Br$_2$ as the internal standard; Z/E ratios are given within parentheses.
[c] Solvent (0.25 mL).
[d] Solvent free.

significantly improved yield, and **3aa** could be formed in 79% yield with an excellent Z/E stereoselectivity of 20:1 under solvent-free conditions. (entry 10) Remarkably, to the best of our knowledge, transition metal-catalyzed native carboxylic acid-directed highly stereoselective C–H activation allylation was unsuccessful until we reported here.

**Substrate scope**. With the optimal conditions in hand, the generality of Rh (III)-catalyzed solvent-free C–H activation allylation was first investigated. As illustrated in Fig. 2, *ortho*-substituted benzoic acids regardless of electron-donating and electron-withdrawing groups were amenable to the developed protocol, thus affording **3aa**, **3ca**, **3da**, and **3ea** in moderate to good yields. In contrast, *meta*- or *para*-substituted benzoic acids did not generate monoallylation products but exclusively gave the diallylation products (**3fa-3ha**) except for **3ba**. It is worth mentioning that the indole carboxylic acids were also effective for this transformation. Interestingly, when pyrrole-2-carboxylic acid was chosen as a coupling partner, an unexpected domino C–H activation allylation/decarboxylation product **3ka** was formed. Compared to aryl carboxylic acids-directed C–H activation allylation, α, β-unsaturated carboxylic acids-directed vinylic C–H bond activation allylation was rarely reported due to the higher bond dissociation energy (BDE). Gratifyingly, various α, β-unsaturated carboxylic acids could smoothly react with **2a**, furnishing the desired products in satisfactory yields and high stereoselectivities (**3na-3pa**). Furthermore, a range of vinylethylene carbonates (VECs) bearing electron-donating as well as electron-withdrawing groups at the *para-*, *meta-*, *ortho*-position of the aryl (Supplementary Fig. 3) could perform well and deliver the corresponding allylic alcohols in good yields with excellent stereoselectivities (**3ab-3aj**). Additionally, the halogen substituents, such as bromo (**3ag**), chloro (**3aj**), fluoro (**3af** and **3ai**), were well-tolerated to the standard conditions, which could be utilized in orthogonal cross couplings for further structural elaboration. Notably, except alkyl groups, the tolerance of functional groups

on VECs could be extended from simple aryl to heteroaryl without any stereoselectivity erosion (**3al** and **3am**). The late-stage functionalization of Repaglinide which is an antidiabetic drug was successfully accomplished by introducing allylic alcohols in a decent yield.

**Mechanistic studies and synthetic applications**. After having demonstrated a broad scope of the reaction, our attention turned to explore the mechanism of this Rh-catalyzed[18,20–23] weakly coordinating directed C–H functionalization[24–26] (Fig. 3). Initially, Rh (III)-catalyzed C–H activation allylation reaction was carried out in the presence of isotopically labeled D$_2$O, and a significant 75% deuterium was incorporated at the ortho-position of **1a** (Supplementary Fig. 6). It clearly revealed that the formation of rhodacycle was a reversible step. Besides, a parallel kinetic experiment with substrate **1c** and its isotopically labeled D-**1c** showed a kinetic isotope effect (KIE) of $k_H/k_D$ 2.0 (Supplementary Figs. 7, 8), which indicated that the C–H activation could be turnover-limiting step[27].

We further evaluated the robustness of Rh (III)-catalyzed C–H activation allylation reaction by performing a large-scale experiment and an array of derivatizations (Fig. 4). The C–C bond formation reaction was efficient even on 5 mmol scales. Moreover, the double bond of **3a** could also be oxidized under VO (acac)$_2$ and *t*BuOOH condition to give an epoxidation product **5a** in 57% isolated yield. Treatment of **3a** under Pd/C and H$_2$ reaction condition, an aliphatic alcohol **6a** could be formed in a respectable yield.

On the basis of above mechanistic studies and previous reports[15,19], we propose a tentative mechanism for this Rh (III)-catalyzed C–H activation allylation reaction (Fig. 5). The catalytically active species [Cp*Rh](SbF$_6$)$_2$ is first generated by a counteranion exchange from [Cp*RhCl$_2$]$_2$ with AgSbF$_6$. After that, a reversible concerted metalation deprotonation (CMD)[28,29] (Supplementary Fig. 9, Supplementary Data 1) takes place under the direction of carboxylate to give a 5-membered rhodacycle (**II**),

**Fig. 2 Substrate scope of Rh(III)-catalyzed solvent-free C–H activation allylation.** [a]Reaction conditions: **1a** (0.1 mmol), **2a** (0.15 mmol), [Cp*RhCl$_2$]$_2$ (5 mol %), AgSbF$_6$ (20 mol%), at 60 °C for 12 h, After the reaction was complete,1 mL DMF, 0.15 mmol K$_2$CO$_3$ and 0.3 mmol MeI were added and the mixture was stirred at rt for 3 h, isolated yields of corresponding methyl esters, Z/E ratios in parentheses (Supplementary Figs. 4, 5). [b]0.3 mmol **2a**. [c]36 h.

**Fig. 3 Mechanistic studies. a** H/D exchange experiment. **b** The parallel experiments of kinetic isotope effect (KIE).

which could coordinate to **2a** followed by migratory insertion/β-oxygen elimination to yield intermediate (**III**). Finally, protonation of **III** provides C–C bond formation product **3a** together with Rh species for the next cycle.

**(Z)-allylic-supported macrolides construction.** Since the crucial connection of the carboxylic acids and vinylethylene carbonates building blocks was successfully established, we next selected

**Fig. 4 Synthetic applications of this method. a** Gram-scale reaction of **3a**. **b** Oxidation and reduction of **3a**.

**Fig. 5 Proposed mechanism.** This proposed mechanism consists of concerted metalation deprotonation (CMD), migratory insertion/$\beta$-oxygen elimination and protonation.

different natural amino acids as building blocks and attempted to assemble them to the macrolides. Generally, a series of aromatic, heteroaromatic, and $\alpha$, $\beta$-unsaturated motifs containing (Z)-allylic-supported or derivative macrolides were obtained in moderate isolated yields after three steps by a biomimetic modularization strategy (Fig. 6). Apart from the 14-membered macrolides construction, we also applied these linkers to staple tripeptide, and a 17-membered macrolide was successfully provided in a synthetically useful yield. Additionally, the antidiabetic drug Repaglinide bearing allylic alcohol linker could efficiently react with dipeptide, which demonstrated the potential of biomimetic modularization strategy to achieve drug-like macrolide. Furthermore, alkyl alcohols derived from allylic alcohols by reduction proved to be viable linkers as well, enabling facile access to the less rigid macrolide, which highlighted allylic alcohols were multifunctional linkers.

**Reversal of MDR mediated by P-gp.** To explore whether these allylic-supported macrolides could successfully transfer pharmacologically relevant features or not, these collected compounds were screened in different test systems and they exhibited excellent potency as an inhibitor of P-gp-mediated MDR in tumor cell lines. These compounds on reversing MDR to chemotherapeutic agents were evaluated using human oral epidermoid carcinoma KB, human leukemia K562 cells and their drug resistant counterparts KBV200 and K562/ADR cells with P-gp overexpression. **4b**, **4f**, **4g**, **4h**, at the non-cytotoxic concentration of 10 μM significantly restored the sensitivities of KBV200 and K562/ADR cells, but not KB or K562 cells, to cytotoxic agents vinorelbine and docetaxel, two P-gp substrates, with the fold-reversals ranging from 1.8 to 176.3 (Table 2, Fig. 7a), suggesting that the reversing activity of these compounds may be through inhibiting P-gp functions. Moreover, it is noteworthy that at the same concentration of 10 μM, the activity of **4g**, with fold-reversals from 89.3 to 176.3, is much more potent than that of the first-generation P-gp inhibitor verapamil (fold-reversals from 26.8 to 67.2) (Table 2).

P-gp inhibitors reverse MDR either through down-regulating P-gp expression or through inhibiting the efflux function of P-gp[30]. As shown in Figs. 7, **4b**, **4f**, **4g**, and **4h** did not affect P-gp expression in KBV200 cells (Fig. 7b, Supplementary Fig. 1), while increased the accumulation of rhodamin-123 (Rho-123), a P-gp substrate, and inhibited the efflux of Rho-123 in KBV200 cells, but not in KB cells (Fig. 7c, d and Supplementary Fig. 2). Their activities on Rho-123 uptake and efflux were consistent with the results from cytotoxicity assays, which indicated that **4b**, **4f**, **4g**, and **4h** exhibited their activities by inhibiting the function of P-gp transporter, rather than by altering P-gp expression. Furthermore, the activities of **4g** on both increasing the Rho-123 accumulation and inhibiting Rho-123 efflux in KBV200 cells were much more potent than that of verapamil. Collectively, these results indicated that **4b**, **4f**, **4g**, and **4h** inhibit P-gp transporter function, thus reversing MDR.

## Discussion

In summary, a biomimetic modularization strategy to create macrolides through native carboxylic acids directed Rh (III)-catalyzed C–H activation allylation is described. This synthetic methodology features mild conditions, broad substrate scope, and high chemo- and stereoselectivity. Notably, the concise biomimetic strategy not only surmounts the difficulty of gram-scale resupply from natural product, but also avoids the requirement of multistep total synthesis. Moreover, the functionality of these (Z)-allylic-supported macrolides was highlighted by fighting P-gp-mediated MDR in cancer chemotherapy with 180 fold-reversals. Given these promising results, we believe our biomimetic modularization strategy will be a valuable resource for the exploration of functional macrocyclic compounds chemical space.

## Methods

**General information.** NMR spectra were recorded at room temperature on the following spectrometers: Bruker Avance III 400 Spectrometer (400 MHz), Bruker Avance III 500 (Cryo) Spectrometer (500 MHz), and Bruker Avance III 600 Spectrometer (600 MHz). Chemical shifts are given in ppm and coupling constants in Hz. [1]H spectra were calibrated in relation to the reference measurement of TMS (0.00 ppm). [13]C spectra were calibrated in relation to deuterated solvents. The

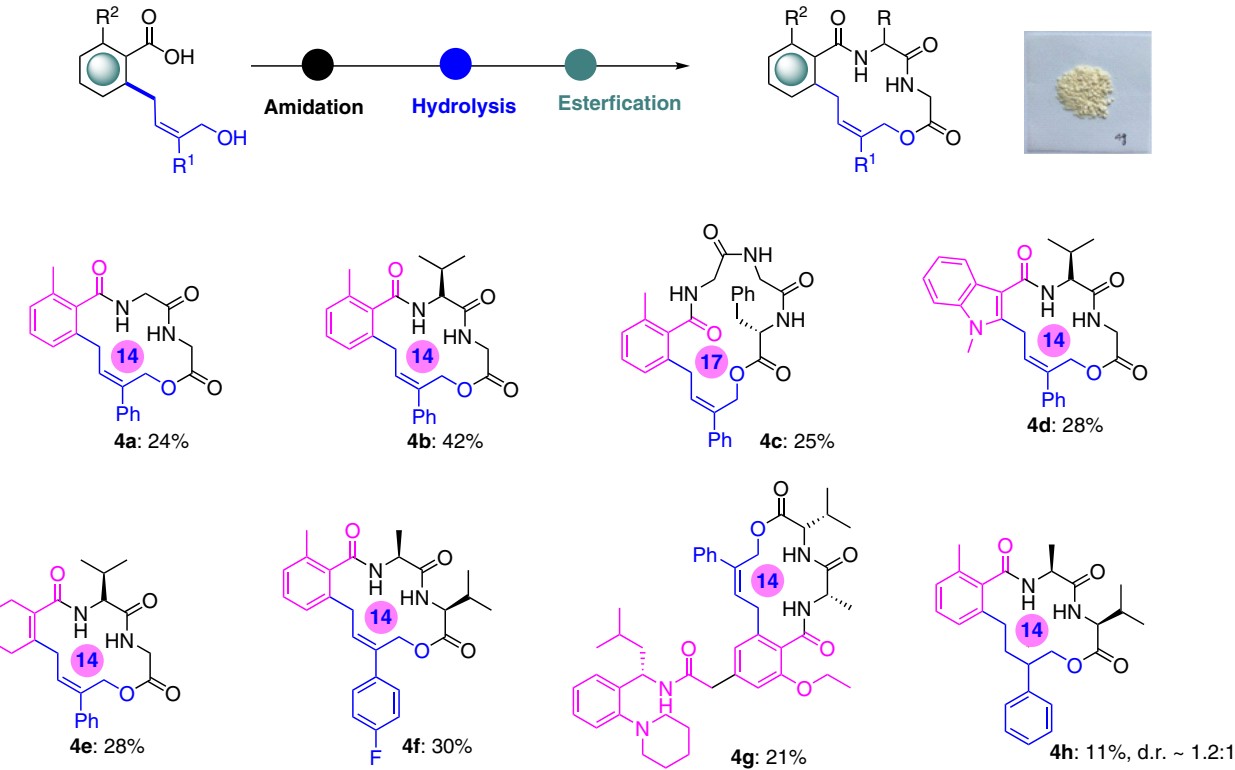

**Fig. 6 Construction of functional (Z)-allylic-supported macrolides by a biomimetic modularization strategy.** [a]Reaction conditions: i) NH₂-A-A-COOMe (1.5 equiv.), EDCI (1.2 equiv.), HOBt (1.5 equiv.), Et₃N (4.5 equiv.), DMF, r.t., overnight or NH₂-A-A-COOMe (1.5 equiv.), HATU (1.5 equiv.), DIPEA (4.0 equiv.), DMF, r.t., overnight; ii) LiOH (10.0 equiv.), MeOH/H₂O (1:1), r.t., 1 h; iii) EDCI (3.0 equiv.), HOBt (3.0 equiv.), DMF/DCM (1:4), r.t., overnight. [b]The picture is 500 mg of **4g** synthesized by this method.

**Table 2 Reversal effects of compounds (10 μM) on P-gp-mediated MDR.**

| Drugs | IC₅₀ (nM, mean ± SD) | |
|---|---|---|
| | **KBV200** | **K562/ADR** |
| vinorelbine | 433.6 ± 9.1 | 557.6 ± 19.3 |
| vinorelbine+**4b** | 108.5 ± 0.9 (4.0) [a] | 189.1 ± 28.5 (2.9) [a] |
| vinorelbine+**4f** | 18.7 ± 0.4 (23.2) [a] | 30.1 ± 1.5 (18.5) [a] |
| vinorelbine+**4g** | 3.8 ± 0.1 (113.0) [a] | 6.2 ± 1.2 (89.3) [a] |
| vinorelbine+**4h** | 52.0 ± 0.3 (8.3) [a] | 71.5 ± 3.2 (7.8) [a] |
| vinorelbine+verapamil | 12.3 ± 0.1 (35.4) [a] | 20.8 ± 2.9 (26.8) [a] |
| docetaxel | 103.7 ± 4.8 | 1029.5 ± 20.5 |
| docetaxel+**4b** | 20.6 ± 0.8 (5.0) [a] | 557.9 ± 146.4 (1.8) [a] |
| docetaxel+**4f** | 2.6 ± 0.1 (39.3) [a] | 29.7 ± 6.0 (34.6) [a] |
| docetaxel+**4g** | 0.6 ± 0.0 (170.8) [a] | 5.8 ± 1.4 (176.3) [a] |
| docetaxel+**4h** | 5.6 ± 0.1 (18.5) [a] | 97.2 ± 9.7 (10.6) [a] |
| docetaxel+verapamil | 1.5 ± 0.0 (67.2) [a] | 25.0 ± 0.6 (41.1) [a] |

[a]The fold-reversal of MDR was defined as the ratio of the IC₅₀ of the cytotoxic agents in the absence of compounds to that in the presence of compounds (10 μM). Cytotoxicity was determined by sulforhodamine B (SRB; KBV200 cells) or (3-(4,5-Dimethylthiazol-2-yl)-2,5-diphenyltetrazolium bromide (MTT; K562/ADR cells) assay as described in Methods. Data are presented as means ± SD of three independent experiments.

uniformly, otherwise it could affect the Z/E ratio and yield, then stirred at 60 °C for 12 h. After that, the vial was cooled to r.t., added K₂CO₃ (0.3 mmol, 3.0 equiv.) and MeI (0.6 mmol, 0.6 equiv.), then diluted with DMF (1 mL). The mixture was stirred at r.t. for 3 h, and then quenched with saturated brine and extracted with ethyl acetate. The combined organic layers were dried over anhydrous Na₂SO₄, concentrated in vacuo to give the residue. The ratio of Z/E-isomers were determined by ¹H NMR analysis of crude reaction mixture. The crude residue was purified by FCC (PE/EA = 3:1, Rf = 0.2–0.4) to get the target product as light oil.

**Cell culture.** The human oral epidermoid carcinoma KB and its P-gp-overexpressing counterpart KBV200 cell lines were kindly provided by Professor Liwu Fu (Yat-sen University, Guangzhou), the K562 cell line was purchased from the American Type Culture Collection (ATCC; Manassas, VA, USA), the P-gp-overexpressing K562/ADR cell line was kindly provided by Professor Lisa Oliver (INSERM, Nantes, France).Cells were maintained in RPMI-1640 medium containing 10% fetal bovine serum (FBS) at 37 °C in a humidified 5% CO₂ atmosphere.

**Cell proliferation assay.** Cytotoxicity was determined using sulforhodamine B (SRB) or (3-(4,5-Dimethylthiazol-2-yl)-2,5-diphenyltetrazolium bromide (MTT) assay. Cells were grown in 96-well plates and treated with various concentrations of compounds, alone or in combination as indicated, for 72 h. For KB and KBV200 cells, cells were stained with SRB for 30 min at room temperature and bound SRB was solubilized with 10 mM Tris and absorbance was measured at 510 nm. For K562 and K562/ADR cells, cells were stained with MTT for 4 h and absorbance was measured at 570 nm. The IC₅₀ (concentration required for 50% inhibition) was calculated by the Logit method. Fold resistance for individual compounds was defined as the IC₅₀ of MDR cancer cells divided by that of the parental drug-sensitive cells.

**Drug uptake and efflux assay.** For Rho-123 uptake assay, cells were incubated with 5 μM Rho-123, in the presence or absence of compounds at 37 °C for 1 h. For Rho-123 efflux assay, cells were incubated with 10 μM Rho-123 for 1 h, and then washed, incubated in Rho-123-free medium, with or without compounds for additional 1 h. Cells were photographed under an Olympus fluorescence microscope (Olympus Optical, Tokyo, Japan) or analyzed with an ACCURI C6 PLUS instrument (BD Biosciences, Billerica, MA, USA).

following abbreviations were used for ¹H NMR spectra to indicate the signal multiplicity: s (singlet), d (doublet), t (triplet), q (quartet), and m (multiplet) as well as combinations of them. For HRMS data, the ESI-positive method was applied on the Agilent G6520 Q-TOF, or EI method was applied on the Themo Fisher Scientific Thermo DFS. Chemicals were purchased from commercial suppliers. Unless stated otherwise, all the substrates and solvents were purified and dried according to standard methods prior to use.

**General procedure for C–H activation.** A teflon-capped vial was charged with the respective benzoic acid (0.1 mmol, 1.0 equiv.), vinylethylene carbonate (0.15 mmol, 1.5 equiv.), K₂CO₃ (0.05 mmol, 0.5 equiv.), [Cp*RhCl₂]₂ (5 mol%), AgSbF₆ (20 mol%) under an air atmosphere. The reaction mixture was mixed

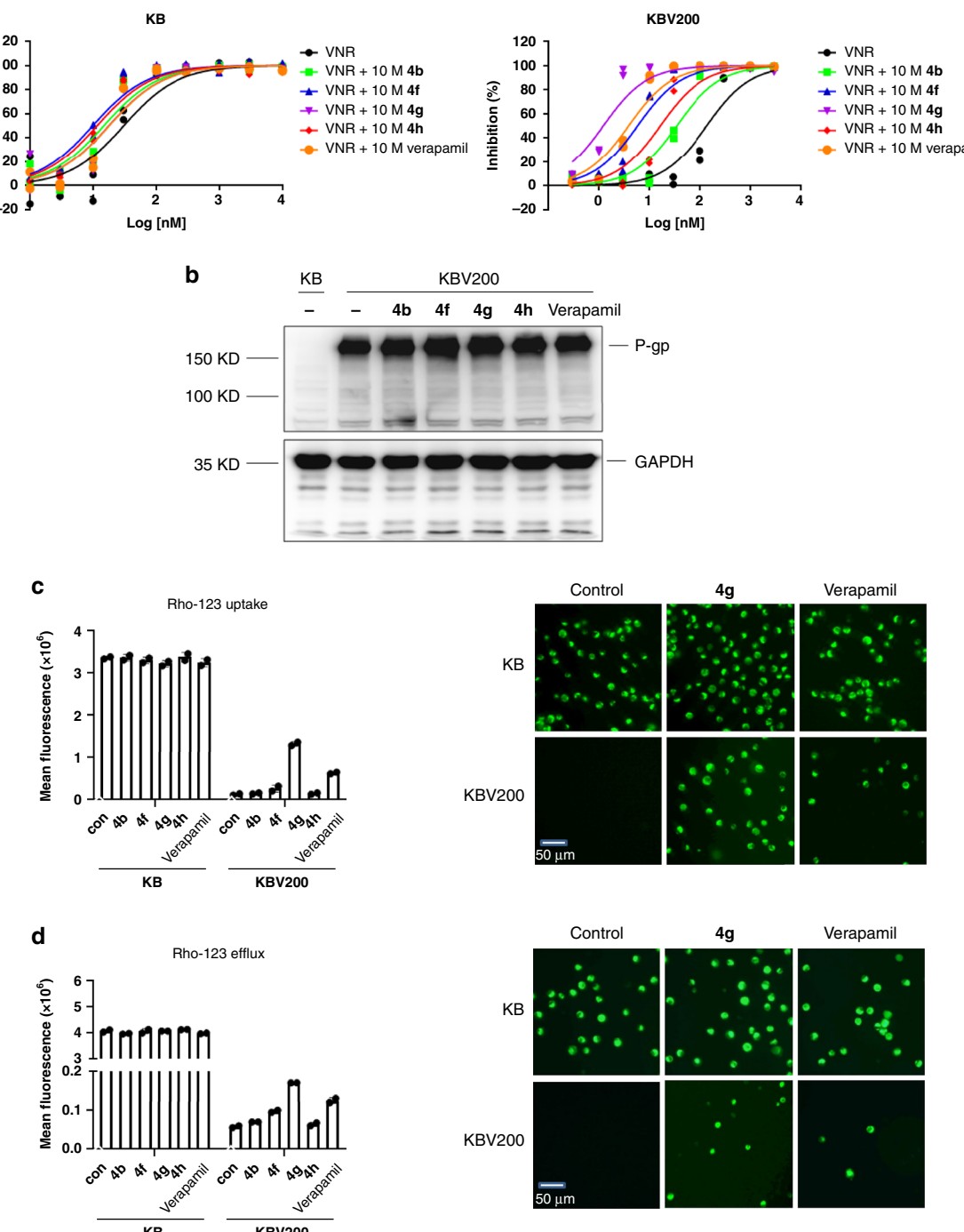

**Fig. 7 Reversal of MDR mediated by P-gp. a** KB or KBV200 cells were treated with vinorelbine (VNR) in the absence (black circles) or presence of 10 μM **4b** (green squares), **4f** (blue triangles), **4g** (purple downward triangles), **4h** (red diamonds) or verapamil (orange circles) for 72 h, and cytotoxicity was determined by sulforhodamine B assay, n = 2 independent experiments. **b** KB or KBV200 cells were treated with 10 μM **4b**, **4f**, **4g**, and **4h** or verapamil for 24 h, and P-gp and GAPDH were analyzed by Western blotting. **c** KB or KBV200 cells were incubated with 5 μM rhodamin-123 (Rho-123), in the presence or absence of 10 μM compounds at 37 °C for 1 h, and Rho-123 accumulation was determined by fluorescence microscope or analyzed by flow cytometry. **d** KB or KBV200 cells were incubated with 10 μM Rho-123 for 1 h, and then washed, incubated in Rho-123-free medium, with or without 10 μM compounds for additional 1 h, and efflux of Rho-123 were determined by fluorescence microscope or analyzed by flow cytometry. A representative result is shown from two independent experiments.

**Western blotting**. After drug treatment for 24 h, cells were washed twice with cold phosphate-buffered saline (PBS; 137 mM NaCl, 2.7 mM KCl, 10 mM Na$_2$HPO$_4$, and 1.8 mM KH$_2$PO$_4$, pH 7.4), lysed in sodium dodecyl sulfate (SDS) sample buffer, and boiled for 10 minutes. Cell lysates containing equal amounts of protein were separated by SDS-PAGE and transferred to PVDF membranes (Millipore, Bedford, MA, USA). After blocking in 5% nonfat milk in TBST (Tris-buffered saline containing 0.1%

Tween-20, pH 7.6), membranes were incubated with anti-P-gp antidody (sc-1517, Santa Cruz) and anti-GAPDH antidody (60004-1-Ig, Proteintech) with a dilution of 1:1000 at 4 °C overnight and then exposed to appropriate secondary antibodies with a dilution of 1:1000 for 2 h at room temperature. Immunoreactive proteins were visualized using the enhanced chemiluminescence system from Pierce Chemical (Rockford, IL, USA). Uncropped blots were shown in Supplementary Fig. 1.

**Reporting summary.** Further information on research design is available in the Nature Research Reporting Summary linked to this article.

## Data availability
All relevant data are available in Supplementary Information, Supplementary Data and from the authors.

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

## Acknowledgements
The authors gratefully acknowledge 100 talent program of Chinese Academy of Sciences, NSFC (21702217), "1000-Youth Talents Plan", Shanghai-Youth Talent, National Science & Technology Major Project" Key New Drug Creation and Manufacturing Program" China (Number: 2018ZX09711002-006), Shanghai-Technology Innovation Action Plan (18JC1415300), and the Science and Technology Commission of Shanghai Municipality (Number: 18DZ2293200).

## Author contributions
L.C. designed and carried out most of the chemical reactions and analyzed the data. H.Q. performed the biological experiments. Z.X., H.W. and Y.X. supported the performance of synthetic experiments. W.Y. and L.L. designed the experiments. W.Y. conceived the idea and supervised the research. W.Y. and L.C. prepared the manuscript and supporting information.

## Competing interests
The authors declare no competing interests.
