## [Peer Review File · Nature Communications]

Reviewers' comments:

Reviewer #1 (Remarks to the Author):

In this work, Yang and Luo reported a synthesis of macrolides and their biological activities toward multidrug resistant cancer. In the synthesis part, the key reaction is the Rh(III)-catalyzed C-H carboxylic acid-directed allylation with vinyl cyclic carbonates (VCCs), providing an allyl alcohol. By follow-up amidation, hydrolysis and esterification reaction, this product could be elaborated to macrolides. A close look at the literature (two reviews cited in ref 28 and 33) showed that VCCs have been previously used by Wang, Kim, Ackermann, glorius and Yu already in C-H activation reactions with different metals and directing groups (see ref 43-47 in *Angew. Chem. Int. Ed.* 2018, 57, 13735 – 13747). The authors certainly overlooked these literature precedents and no comments on the specific advances compared to previous work have been made. In my view, the reaction could be regarded as a simple extension of known reactions by using a different directing group. Therefore, I could not see the high novelty that is suitable for publication in *Nat. Commun.*

Reviewer #2 (Remarks to the Author):

Recommendation: Publish after minor revisions noted.

Comments:

This manuscript from Yang et al describes a synthetic methodology for the synthesis of poly-substituted allylic alcohol through native carboxylic acids directed Rh (III)-catalyzed C-H activation allylation. The authors exemplified their protocol over a large number of examples, and provided unprecedentedly rapid access to various (Z)-allylic-supported macrolides from the generated poly-substituted allylic alcohol. Furthermore, on the basis of some mechanistic studies, the authors propose a tentative mechanism for this Rh (III)-catalyzed C-H activation allylation reaction. Notably, the functionality of these novel (Z)-allylic-supported macrolides was highlighted by fighting P-gp-mediated multidrug resistance (MDR). In general, this is an interesting investigation, I think that the synthetic strategy will be a valuable resource for the exploration of functional macrocyclic compounds chemical space. The manuscript describes the performed research in a concise and clear manner. The syntheses are very efficient and the biological evaluation has been well performed. Plenty of data are presented, and the study is logical. Thus, the manuscript is sufficiently important to be published in *Nature Communications*. Nevertheless, the authors should attempt to address the following points to further enhance the value of the manuscript:

1. The carboxylic acids were used as the starting material to react with peptide to construct the macrolides, but in the C-H activation allylation reaction they performed in the methodology study were carboxylates, why?
2. How to determine the ratios of Z/E-isomers by ¹H NMR analysis of the crude reaction mixture or by the isolated yields? If they were determined by ¹H NMR analysis of the crude reaction mixture, the ¹H NMR spectroscopy of crude reaction mixture of the optimal condition should be provided in Supporting Information.
3. The authors claim that: "the tolerance of functional groups on VECs could be extended from simple aryl to heteroaryl without any stereo selectivity erosion (3al and 3am)". I want to know how about the alkyl groups?
4. In Table 2, note c is missing.
5. "eq" should be changed to "equiv" in Supporting Information. "eq" = equation; "equiv" = equivalent.

Reviewer #3 (Remarks to the Author):

The manuscript from the groups of Lou and Yang reports the development of a method for C-H

allylation, its application to the synthesis of novel macrolides and their evaluation as inhibitors of Pgp-mediated multidrug resistance (MDR) in tumor cell lines. The Rh-catalyzed carboxylic acid-directed C-H allylation is, to the best of my knowledge, previously unreported and thereby of large interest to the scientific community. This is also the case for the reversal of multidrug resistance, whereas the transformation of the C-allylated intermediates to macrolides can be regarded as a bit more routine, keeping in mind that macrocyclizations are never trivial. However, as described below the manuscript has several major and many minor shortcomings that must be addressed before it can be considered for publication. Therefore I recommend that the current version of the manuscript is rejected, but I suggest that a thoroughly revised version can be submitted at a later stage.

General comments

Major revisions

1. It is not evident why the set of macrolides were designed, synthesized and evaluated as inhibitors of MDR. Were they designed based on similarity to known MDR inhibitors, or made as part of an effort to prepare compound collections for screening in different test systems? This should be clarified.
2. The manuscript is in need of linguistic revision. Several parts are difficult to understand, or make little sense. The first sentence in the second paragraph of the Introduction which reads "While the traditional total synthesis or semisynthetic approaches¹⁶, which could modify natural macrocyclic molecules on some unique reaction sites, may not be effective anymore when the multidrug-resistance has occurred." is one example.
3. Throughout the manuscript the synthesis of the macrolides is described as being done by a "biomimetic strategy" and building blocks are referred to as being biomimetic. The authors must clarify in what way their strategy is biomimetic, and why building blocks are described as being biomimetic, if these descriptions are to be kept in a future version of the manuscript. For instance, is there a similarity to known biosynthetic pathways? All compounds that we prepare as chemists are made by assembly of different building blocks, but this does not translate to that all synthetic strategies are biomimetic. The route described in this manuscript is novel and there is no need to try to hype it by describing it as being biomimetic if this is not correct.

Introduction

Major revision

- 1) The first paragraph and Figure 1A describes the assembly of proteins and oligosaccharides from building blocks. The text and Figure 1B then describe that macrocycles can be assembled from building blocks. As building blocks possess structural information it is concluded that the macrocycles could show multifunctional biological activities. This is logic is far from being correct, and should be removed from the manuscript. There is no reason to try to link proteins and oligosaccharides to macrocycles for readers of Nature Communications. Please remove this somewhat naive connection.
- 2) Figure 1. This figure lacks a proper legend that described the different panels. This should be included. Why have results from the testing of the macrolides prepared in the manuscript been included as panel D. This should be moved to the Results and Discussion part. If panel B is to be kept there must be a reason which is connected to the rest of the manuscript. Currently it shows four macrocycles in different colors without any explanation.

Results and Discussion: Synthetic part

The presented Rh-catalyzed C-H functionalization is novel, but it is not entirely surprising that it works since amide-directed allylations have been reported (Org. Lett. 2013, 15, 17, 4576). This should be mentioned in the manuscript. The scoping of the allylation is nicely done and the late stage functionalization of repaglinide is a very good example of the power of the method reported in the manuscript. The use of an allylation as a synthetic tactic in this context is novel.

Major revisions

1. The authors fail to cite large areas of precedent in Rh-catalyzed C-H functionalization. Major

works by Glorius, Davies, Saá, MacGregor and several others should be cited on Rh chemistry. For concerted metalation demethylation CMD mechanisms, the work of Davies (Chem. Rev. 2017, 8649), Fagnou and others should be cited, as should Ackermann's review on the role of carboxylates in C-H activation (Chem. Rev.) Ackermann and Jin-Quan Yu's various studies on weakly coordinating DGs is missing. In summary, the citation of previous studies in the field is very poor in the manuscript.

2. The text says: "a significant 75% deuterium was incorporated at the ortho-position of 1a. It clearly revealed that the formation of rhodacycle via concerted metalation deprotonation (CMD) was a reversible step." This experiment establishes reversibility, but not that it is a CMD mechanism, and the authors should be clear about that.

3. The kinetic isotope effect (KIE) is determined from one experiment. This is not adequate. For a discussion, please read and cite Hartwig's review of using KIEs in this way: Angew. Chem. Int. Ed., 2012, 3066. More experiments are required to support the proposal.

4. The KIE value of 10.5 is far above a believable level. It is considered that KIE of 6.9 is the theoretical maximum. Rh- and Pd-catalyzed CMD usually has values of 3-5, which are "high". For context, a related Rh-catalyzed C-H allylation reaction reported by Saá finds a KIE of 3.5 (Org. Lett. 2013, 15, 17, 4576). The authors should thoroughly re-examine the accuracy of their KIE experiments, do additional (e.g. internal competition) experiments, and refer to the above Hartwig discussion.

5. If the additional/revised experiments support CMD, Scheme 3 should include this. This would include carboxylate species formed from the initial dicationic Rh complex and a CMD transition state. Currently, the mechanistic cycle is silent on the C-H activation, which makes it rather uninformative. It is likely that any dicationic Rh species formed would encounter and coordinate a carboxylate in the reaction mixture, or even two. One carboxylate could be from the substrate undergoing C-H activation; the other could participate in the CMD transition state.

6. The first on-cycle complex is written as $1/2[\text{Cp}^*\text{Rh}(\text{SbF}_6)_2]_2$. This is very unlikely as SbF_6^- is very weakly coordinating and unable to bridge two Rh centres. The complex would be better written as $[\text{Cp}^*\text{Rh}](\text{SbF}_6)_2$.

7. The formation of HSbF_6 in the mechanism is overwhelmingly unlikely. HSbF_6 is a super-acid and would never form under these conditions. Also, 1 equiv of acid is enough to protonate both the carboxylate in II and the O-Rh bond, so the cycle is not drawn balanced (Scheme 3). In addition, what is the fate of the K cation? The authors should balance the incoming/outgoing species.

Minor revisions

1. The Rh-catalyzed allylation should not be called a "Tsuji-Trost" type reaction. It's Rh, not Pd-, catalysed, it is not proposed to proceed via η^3 allyl intermediates and the aryl group is not really a nucleophile. Also, the Rh(III) centre would have to be oxidized to Rh(V) during the cycle, which is highly unlikely here, and which the authors don't propose.

2. Were aldehyde products detected from the Rh-catalysed oxidation of alcohols 3 (Scheme 3)? Aliphatic alcohols might undergo beta-hydride elimination from alkoxide species such as II.

3. The text would benefit from a clarification about the "solvent-free" conditions. Are all the reagents in the mixture solid throughout, or does one of them effectively act as a solvent?

4. The top part of Scheme 4 would benefit from inclusion of experimental detail about the steps in the conversion of alcohols 3 to macrolides, e.g. was the dipeptide moiety incorporated as a dipeptide building block (which reagents were used) and what were the conditions that were used for macrocyclization? What is the small picture inserted at the top right in this figure? Further details should go into the legend.

5. Scheme 1 has the word "standard" misspelled above the second reaction arrow.

6. Scheme 3 is not actually a Scheme. It should be a Figure with a legend.

Results and Discussion: Pharmacology part

Major revisions

1. Macrolide 4g, which is a derivative of repaglinide, was used to determine that the compounds inhibit the activity of Pgp, not Pgp-expression. However, the structure of 4g is very different from

that of 4b, 4f and 4h, which also reduce Pgp mediated MDR. It is therefore possible that macrolide 4g could have a different mode of action than 4b, 4f and 4h. Therefore the mode of action of one of 4b, 4f and 4h should be investigated in the same way as for 4g.

Reviewer: 1

Comments:

In this work, Yang and Luo reported a synthesis of macrolides and their biological activities toward multidrug resistant cancer. In the synthesis part, the key reaction is the Rh(III)-catalyzed C-H carboxylic acid-directed allylation with vinyl cyclic carbonates (VCCs), providing an allyl alcohol. By follow-up amidation, hydrolysis and esterification reaction, this product could be elaborated to macrolides. A close look at the literature (two reviews cited in ref 28 and 33) showed that VCCs have been previously used by Wang, Kim, Ackermann, glorius and Yu already in C-H activation reactions with different metals and directing groups(see ref 43-47 in Angew. Chem. Int. Ed. 2018, 57, 13735 – 13747). The authors certainly overlooked these literature precedents and no comments on the specific advances compared to previous work have been made. In my view, the reaction could be regarded as a simple extension of known reactions by using a different directing group. Therefore, I could not see the high novelty that is suitable for publication in Nat. Commun..

Response: The advances of our chemistry are obvious. First, we bridged the gap between Rh-catalyzed C-H allylation and bioactive macrocyclic molecules via a short and modular biomimetic strategy. It should be noted that there are many Rh-catalyzed C-H activation reactions, but the applications are very rare. Second, our reaction can control the stereoselectivity of poly-substituted allylic alcohols which can be utilized as multifunctional linkers to construct macrocycles. In this case, they are not accessible via the previous known reactions.

Reviewer: 2

Comments:

This manuscript from Yang et al describes a synthetic methodology for the synthesis of poly-substituted allylic alcohol through native carboxylic acids directed Rh (III)-catalyzed C-H activation allylation. The authors exemplified their protocol over a large number of examples, and provided unprecedentedly rapid access to various (Z)-allylic-supported macrolides from the generated poly-substituted allylic alcohol. Furthermore, on the basis of some mechanistic studies, the authors propose a tentative mechanism for this Rh (III)-catalyzed C-H activation allylation reaction. Notably, the functionality of these novel (Z)-allylic-supported macrolides was

highlighted by fighting P-gp-mediated multidrug resistance (MDR). In general, this is an interesting investigation, I think that the synthetic strategy will be a valuable resource for the exploration of functional macrocyclic compounds chemical space. The manuscript describes the performed research in a concise and clear manner. The syntheses are very efficient and the biological evaluation has been well performed. Plenty of data are presented, and the study is logical. Thus, the manuscript is sufficiently important to be published in Nature Communications.

The carboxylic acids were used as the starting material to react with peptide to construct the macrolides, but in the C-H activation allylation reaction they performed in the methodology study were carboxylates, why?

Response: The coordination ability of carboxylic acids is relatively weak compared to their carboxylates. In order to address this problem, we usually added some base to in situ generate carboxylate.

How to determine the ratios of Z/E-isomers by ¹H NMR analysis of the crude reaction mixture or by the isolated yields? If they were determined by ¹H NMR analysis of the crude reaction mixture, the ¹H NMR spectroscopy of crude reaction mixture of the optimal condition should be provided in Supporting Information.

Response: The ratio of Z/E was determined by ¹H NMR analysis of the crude product before purification. The ¹H NMR spectroscopy of crude reaction mixtures of optimal condition has been provided in page 6 in the revised Supporting Information.

The authors claim that: "the tolerance of functional groups on VECs could be extended from simple aryl to heteroaryl without any stereo selectivity erosion (3al and 3am)". I want to know how about the alkyl groups?

Response: It is the limitation in the current methodology. The VECs with alkyl groups could not afford any desired products under the optimal condition. We made a statement in line 10 of page 6

in the revised manuscript.

In Table 2, note c is missing.

Response: Sorry for this mistake. We have added it in the revised manuscript.

“eq” should be changed to “equiv” in Supporting Information. “eq” = equation; “equiv” = equivalent.

Response: We have corrected it in the Supporting Information.

Reviewer: 3

Comments:

The manuscript from the groups of Lou and Yang reports the development of a method for C-H allylation, its application to the synthesis of novel macrolides and their evaluation as inhibitors of Pgp-mediated multidrug resistance (MDR) in tumor cell lines. The Rh-catalyzed carboxylic acid-directed C-H allylation is, to the best of my knowledge, previously unreported and thereby of large interest to the scientific community. This is also the case for the reversal of multidrug resistance, whereas the transformation of the C-allylated intermediates to macrolides can be regarded as a bit more routine, keeping in mind that macrocyclizations are never trivial. However, as described below the manuscript has several major and many minor shortcomings that must be addressed before it can be considered for publication. Therefore, I recommend that the current version of the manuscript is rejected, but I suggest that a thoroughly revised version can be submitted at a later stage.

It is not evident why the set of macrolides were designed, synthesized and evaluated as inhibitors of MDR. Were they designed based on similarity to known MDR inhibitors, or made as part of an effort to prepare compound collections for screening in different test systems? This should be clarified.

Response: We designed and synthesized these macrolides by evaluating their bioactivities in different test systems. Interestingly, some of them exhibited excellent potency as an inhibitor of Pgp-mediated multidrug resistance (MDR) in tumor cell lines. We made a statement in page 10 in the revised manuscript.

The manuscript is in need of linguistic revision. Several parts are difficult to understand, or make little sense. The first sentence in the second paragraph of the Introduction which reads " While the traditional total synthesis or semisynthetic approaches¹⁶, which could modify natural macrocyclic molecules on some unique reaction sites, may not be effective anymore when the multidrug-resistance has occurred." is one example.

Response: Thanks a lot for this reviewer's suggestion. We have revised them in the manuscript.

Throughout the manuscript the synthesis of the macrolides is described as being done by a "biomimetic strategy" and building blocks are referred to as being biomimetic. The authors must clarify in what way their strategy is biomimetic, and why building blocks are described as being biomimetic, if these descriptions are to be kept in a future version of the manuscript. For instance, is there a similarity to known biosynthetic pathways? All compounds that we prepare as chemists are made by assembly of different building blocks, but this does not translate to that all synthetic strategies are biomimetic. The route described in this manuscript is novel and there is no need to try to hype it by describing it as being biomimetic if this is not correct.

Response: A biomimetic strategy could be described by two parts: mimicking a biosynthetic pathway or using biomimetic building blocks. In our case, it is a modular biomimetic strategy, which simply uses the fundamental building blocks from living organism's endogenous ligand, such as amino acids. They are different from other exogenous build blocks. Although the exogenous build blocks could also be assembled to many compounds, they are not the fundamental building blocks for biological macromolecules in living organisms.

The first paragraph and Figure 1A describes the assembly of proteins and oligosaccharides from

building blocks. The text and Figure 1B then describe that macrocycles can be assembled from building blocks. As building blocks possess structural information it is concluded that the macrocycles could show multifunctional biological activities. This is logic is far from being correct, and should be removed from the manuscript. There is no reason to try to link proteins and oligosaccharides to macrocycles for readers of Nature Communications. Please remove this somewhat naïve connection.

Response: Thanks for this reviewer's suggestion, we have removed them and rewritten in the manuscript.

Figure 1. This figure lacks a proper legend that described the different panels. This should be included. Why have results from the testing of the macrolides prepared in the manuscript been included as panel D. This should be moved to the Results and Discussion part. If panel B is to be kept there must be a reason which is connected to the rest of the manuscript. Currently it shows four macrocycles in different colors without any explanation.

Response: Thanks for this reviewer's suggestion, we have moved the panel D to the Results and Discussion part. And we made the descriptions of these macrocycles in line 3 of page 2 in the revised manuscript.

The presented Rh-catalyzed C-H functionalization is novel, but it is not entirely surprising that it works since amide-directed allylations have been reported (Org. Lett. 2013, 15, 17, 4576). This should be mentioned in the manuscript.

Response: Thanks for this reviewer's suggestion. We have mentioned it in line 26 of page 2 and cited this literature as Ref 18 in the revised manuscript.

The authors fail to cite large areas of precedent in Rh-catalyzed C-H functionalization. Major works by Glorius, Davies, Saá, MacGregor and several others should be cited on Rh chemistry. For concerted metalation demethylation CMD mechanisms, the work of Davies (Chem. Rev. 2017,

8649), Fagnou and others should be cited, as should Ackermann's review on the role of carboxylates in C-H activation (Chem. Rev.) Ackermann and Jin-Quan Yu's various studies on weakly coordinating DGs is missing. In summary, the citation of previous studies in the field is very poor in the manuscript.

Response: Thanks for this review's suggestion. We have added more related literatures as Ref 18, 20-26 in the revised manuscript.

The text says: "a significant 75% deuterium was incorporated at the ortho-position of 1a. It clearly revealed that the formation of rhodacycle via concerted metalation deprotonation (CMD) was a reversible step." This experiment establishes reversibility, but not that it is a CMD mechanism, and the authors should be clear about that.

Response: Thanks for this reviewer's comments. We have to make an apology for our incorrect description of this experiment. And we have changed the statement in the revised manuscript.

The kinetic isotope effect (KIE) is determined from one experiment. This is not adequate. For a discussion, please read and cite Hartwig's review of using KIEs in this way: *Angew. Chem. Int. Ed.*, 2012, 3066. More experiments are required to support the proposal.

Response: Thanks for this reviewer's comments. Hartwig's review was cited as Ref 27 in the revised manuscript, and we have made 2 runs of the KIE experiments to support the proposal.

The KIE value of 10.5 is far above a believable level. It is considered that KIE of 6.9 is the theoretical maximum. Rh- and Pd-catalyzed CMD usually has values of 3-5, which are "high". For context, a related Rh-catalyzed C-H allylation reaction reported by Saá finds a KIE of 3.5 (*Org. Lett.* 2013, 15, 17, 4576). The authors should thoroughly re-examine the accuracy of their KIE experiments, do additional (e.g. internal competition) experiments, and refer to the above Hartwig discussion.

Response: According to Hartwig's discussion, in our reaction, the KIE experiment was re-examined (Supporting Information) and the KIE value of 2.0 was determined as an average of 2 runs.

If the additional/revised experiments support CMD, Scheme 3 should include this. This would include carboxylate species formed from the initial dicationic Rh complex and a CMD transition state. Currently, the mechanistic cycle is silent on the C-H activation, which makes it rather uninformative. It is likely that any dicationic Rh species formed would encounter and coordinate a carboxylate in the reaction mixture, or even two. One carboxylate could be from the substrate undergoing C-H activation; the other could participate in the CMD transition state.

Response: Thanks for this reviewer's suggestion. We have revised the mechanistic cycle in Figure 2, in which the CMD transition state is included as suggested by the reviewer. In the transition state a bicarbonate is involved for C-H deprotonation. This is supported by a preliminary DFT calculations as given in the Supporting Information.

The first on-cycle complex is written as $1/2[\text{Cp}^*\text{Rh}(\text{SbF}_6)_2]_2$. This is very unlikely as SbF_6 is very weakly coordinating and unable to bridge two Rh centres. The complex would be better written as $[\text{Cp}^*\text{Rh}](\text{SbF}_6)_2$.

Response: Thanks for this reviewer's suggestion. We have changed it in the revised manuscript.

The formation of HSbF_6 in the mechanism is overwhelmingly unlikely. HSbF_6 is a super-acid and would never form under these conditions. Also, 1 equiv of acid is enough to protonate both the carboxylate in II and the O-Rh bond, so the cycle is not drawn balanced (Scheme 3). In addition, what is the fate of the K cation? The authors should balance the incoming/outgoing species.

Response: Sorry for these mistakes. We have corrected all of them in the proposed mechanism (Figure 2) in the revised manuscript.

Minor revisions

The Rh-catalyzed allylation should not be called a “Tsuji-Trost” type reaction. It’s Rh, not Pd-, catalysed, it is not proposed to proceed via η -3 allyl intermediates and the aryl group is not really a nucleophile. Also, the Rh(III) centre would have to be oxidized to Rh(V) during the cycle, which is highly unlikely here, and which the authors don’t propose.

Response: Thanks for this reviewer’s comments. We made an apology for our incorrect statements, and we have deleted them in the revised manuscript.

Were aldehyde products detected from the Rh-catalysed oxidation of alcohols 3 (Scheme 3)? Aliphatic alcohols might undergo beta-hydride elimination from alkoxide species such as II.

Response: Although aliphatic alcohols could undergo beta-hydride elimination, the aldehyde products were not detected in our case.

The text would benefit from a clarification about the “solvent-free” conditions. Are all the reagents in the mixture solid throughout, or does one of them effectively act as a solvent?

Response: Actually, I think this reaction do not need any reagents to act as a solvent, even if 2a is liquid. Because this reaction also worked well when 2e and 2m are solid.

The top part of Scheme 4 would benefit from inclusion of experimental detail about the steps in the conversion of alcohols 3 to macrolides, e.g. was the dipeptide moiety incorporated as a dipeptide building block (which reagents were used) and what were the conditions that were used for macrocyclization? What is the small picture inserted at the top right in this figure? Further details should go into the legend.

Response: Sorry for these mistakes. We have added the details to the legend in the revised manuscript.

Scheme 1 has the word "standard" misspelled above the second reaction arrow.

Response: Sorry for this mistake. We have corrected it in the manuscript.

Scheme 3 is not actually a Scheme. It should be a Figure with a legend.

Response: Thanks for this reviewer's suggestion. We have changed it into Figure 2 in the revised manuscript.

Macrolide 4g, which is a derivative of repaglinide, was used to determine that the compounds inhibit the activity of Pgp, not Pgp-expression. However, the structure of 4g is very different from that of 4b, 4f and 4h, which also reduce Pgp mediated MDR. It is therefore possible that macrolide 4g could have a different mode of action than 4b, 4f and 4h. Therefore the mode of action of one of 4b, 4f and 4h should be investigated in the same way as for 4g.

Response: We did the additional experiments to check the mode of action of these compounds (4b, 4f, 4h and 4g). Although their structures are different, the mode of action is pretty much the same. The only difference is the potency. For more details, please see the revised manuscript.

REVIEWERS' COMMENTS:

Reviewer #2 (Remarks to the Author):

This reviewer has checked the revisions and the author's responses points by points, and the revision requirements made by me in the previous round of review have been satisfactorily addressed. I thus suggested this paper to be published on Nature Communications.

Reviewer #2 (Remarks to the Author):

This reviewer has checked the revisions and the author's responses points by points, and the revision requirements made by me in the previous round of review have been satisfactorily addressed. I thus suggested this paper to be published on Nature Communications.

Response: Thanks for the comments provided by this reviewer.